# Transcriptome Analysis of *Heat Shock Factor C2a* Over-Expressing Wheat Roots Reveals Ferroptosis-like Cell Death in Heat Stress Recovery

**DOI:** 10.3390/ijms24043099

**Published:** 2023-02-04

**Authors:** Sundaravelpandian Kalaipandian, Jonathan Powell, Aneesh Karunakaran, Jiri Stiller, Steve Adkins, Udaykumar Kage, Kemal Kazan, Delphine Fleury

**Affiliations:** 1School of Agriculture and Food Sciences, Gatton, QLD 4343, Australia; 2CSIRO Agriculture and Food, St. Lucia, QLD 4067, Australia; 3School of Agriculture, Food and Wine, University of Adelaide, Glen Osmond, SA 5064, Australia

**Keywords:** climate change, heat stress, heat shock factor, ferroptosis, peroxidases, roots, thermotolerance, wheat

## Abstract

Wheat (*Triticum aestivum* L.) growing areas in many regions of the world are subject to heat waves which are predicted to increase in frequency because of climate change. The engineering of crop plants can be a useful strategy to mitigate heat stress-caused yield losses. Previously, we have shown that heat shock factor subclass C (*TaHsfC2a-B*)-overexpression significantly increased the survival of heat-stressed wheat seedlings. Although previous studies have shown that the overexpression of *Hsf* genes enhanced the survival of plants under heat stress, the molecular mechanisms are largely unknown. To understand the underlying molecular mechanisms involved in this response, a comparative analysis of the root transcriptomes of untransformed control and *TaHsfC2a*-overexpressing wheat lines by RNA-sequencing have been performed. The results of RNA-sequencing indicated that the roots of *TaHsfC2a*-overexpressing wheat seedlings showed lower transcripts of hydrogen peroxide-producing peroxidases, which corresponds to the reduced accumulation of hydrogen peroxide along the roots. In addition, suites of genes from iron transport and nicotianamine-related gene ontology categories showed lower transcript abundance in the roots of *TaHsfC2a*-overexpressing wheat roots than in the untransformed control line following heat stress, which are in accordance with the reduction in iron accumulation in the roots of transgenic plants under heat stress. Overall, these results suggested the existence of ferroptosis-like cell death under heat stress in wheat roots, and that *TaHsfC2a* is a key player in this mechanism. To date, this is the first evidence to show that a *Hsf* gene plays a key role in ferroptosis under heat stress in plants. In future, the role of *Hsf* genes could be further studied on ferroptosis in plants to identify root-based marker genes to screen for heat-tolerant genotypes.

## 1. Introduction

Heat is a major abiotic stress factor that affects crop production in many regions of the world. Depending on the intensity and duration of heat stress and the physiological and developmental status of the plant, most plant species are negatively affected by heat stress. However, heat stress is especially problematic for cool season crops, such as wheat (*Triticum aestivum* L.). Because wheat is a crop that meets the calorie demand of a significant portion of the world’s population, a reduced wheat production caused by heat stress can potentially be a significant threat to global food security. Indeed, according to one estimate, the average cereal yield across the globe declined by 7.6% due to heat stress during extreme weather events that occurred between 1964 and 2007 [1].

In addition to the elevated temperatures currently experienced in many wheat-growing regions, the increased atmospheric concentration of CO_2_ and other greenhouse gases is predicted to cause even more intense heat waves and dramatically alter rainfall patterns [2]. Indeed, the frequency, duration, and severity of episodes with exceptionally high temperatures have already been seen to rise in recent years [3,4]. These unusual climatic changes, which are projected to become more severe in future, have the potential to further threaten global food security by severely limiting crop productivity [1]. Global wheat yield is projected to decline by 4.1 to 6.4% with every 1 °C rise in global temperatures [5]. Deryng et al. [6] predicted that strong heat stress during anthesis could be responsible for up to a 52% reduction from the expected yield of spring wheat by 2080.

Plants respond to heat stress by activating many genes involved in hormone biosynthesis, protein degradation, transcriptional activation and repression, and an unfolded-protein response [7,8,9]. In addition, heat stress induces the production of reactive oxygen species (ROS), which can damage membranes and other cellular systems through lipid peroxidation [10]. Distefano et al. [11] demonstrated that heat stress induces ferroptosis-like cell death in Arabidopsis (*Arabidopsis thaliana* (L.) Heynh.) roots. Ferroptosis is a non-apoptotic form of cell death and is dependent on intracellular iron and ROS [12,13,14,15]. Excess free iron (Fe) ions can generate ROS within the cell [16], resulting in lipid peroxidation and ferroptotic cell death, and the inhibition of this response was shown to have a beneficial effect on heat stress tolerance.

Heat stress also induces the production of heat shock proteins (HSPs) that act as molecular chaperones under stress conditions to maintain cellular homeostasis [17,18]. Heat shock factors (HsfA, HsfB and HsfC) play a key role in the regulation of heat-inducible genes, including *HSPs* [19]. HsfA and HsfB have a central role in the regulation of heat stress-related genes in plants [20]. However, the monocot-specific HsfC2 subclass HSPs have been poorly studied [21].

To date, most studies have considered the detrimental effects of heat stress on above-ground plant parts [22,23], although the roots can also be affected by high temperatures under natural conditions, especially when plant canopies are not dense enough and sunlight can directly contact the soil surface [24]. Given that heat and drought stress often occur simultaneously, maintaining a functional root system, that can adsorb water and nutrients from the soil, could be essential to alleviate the damaging effects of heat and drought stress, especially during the early developmental stages of plant growth. Indeed, reducing root heat stress was found to increase shoot biomass by 33 to 160% and grain yield by 18 to 147% in wheat under field conditions [25]. However, currently very little is known about the molecular responses of plant roots to heat stress. Recently, it has been reported that exogenous trehalose modulates gene expression and alleviates the oxidative damage of high temperature stress in wheat roots [26]. In our previous study, it has been shown that the survival rates of *TaHsfC2a-B*-overexpressing wheat seedlings were significantly improved (90%) relative to those of untransformed plants (15%) after exposure to heat stress (2 h at 43 °C) [27]. In addition, a significant correlation was observed between recovery rates and the average root dry weights of the recovered transgenic lines which were 8.7 and 7.8-fold higher than those of the recovered parental line, suggesting that physiological changes in the roots of transgenic plants may have contributed to the heat stress recovery phenomenon.

The aim of this study was firstly to investigate transcriptional responses of wheat roots to heat stress, and secondly to understand the potential roles of *TaHsfC2a* over-expressing roots in contributing to plant recovery after heat stress that can kill most untransformed control plants [27]. Therefore, this study undertook RNA-seq analyses to identify heat stress-inducible root genes that are potentially associated with heat stress tolerance in wheat. A comparative analysis of untransformed control plants (hereafter called wild-type) and *TaHsfC2a* root transcriptomes indicated that *TaHsfC2a* negatively regulates ROS levels under heat stress in roots through class III peroxidases. In addition, the reduced transcripts of Fe transport genes and nicotianamine synthases caused a lower accumulation of Fe in *TaHsfC2a* roots under heat stress. These results suggested the existence of ferroptosis-like cell death in wheat. To the best of our knowledge, no functional genes have been identified in the ferroptosis-like cell death mechanism seen under heat stress in plants. Therefore, this study is the first report that functionally characterizes a gene in the ferroptosis-like cell death pathway under heat stress in plants. Overall, these results provide new insights into heat stress tolerance in plants.

## 2. Results

### 2.1. Root Transcriptome of Wild-Type Wheat Seedlings under Heat Stress

To identify root transcriptional responses to heat stress, the RNA-seq of wild-type roots with and without heat stress was performed. On average, 90% of the reads were aligned to the reference RefSeq v1.0 (Appendix A). RNA-seq analysis was carried out with log_2_ (Fold Change, FC ≥ 2; *p* < 0.05), unless otherwise stated to identify the genes that are at least twofold induced or repressed with a *p* value of < 0.05. We found that 1666 and 1660 genes were up- and down-regulated, respectively, in wild-type roots after heat treatment (Figure 1; Appendix A).

To gain an insight into the nature of molecular responses to heat, gene ontology (GO) enrichment analysis of the differentially expressed genes (DEGs) in the root tissue was performed, and found that the genes belonging to 46 GO categories were up-regulated under heat stress in wild-type roots (Appendix A). Some of these GO categories include “response to heat, nicotianamine synthase activity, the nicotianamine biosynthetic process, the tricarboxylic acid biosynthetic process, protein folding, response to hydrogen peroxide, ROS, iron ion transport, glutathione transferase activity, DNA binding transcription factor activity, heat shock protein binding, transcriptional repressor activity, and RNA polymerase II proximal promoter sequence-specific DNA binding” (Appendix A). In the down-regulated gene sets, 86 GO categories were over-represented. Some of these categories include “ROS, oxidative stress and hydrogen peroxide-related, cellular oxidant detoxification, peroxidase activity, antioxidant activity, cell proliferation and cell wall-related, cellulose and signalling-related” (Appendix A). In addition, in wild-type roots, 182 and 75 transcription factors (TF) encoding genes were up- and down-regulated, respectively after heat stress (Figure 2; Appendix A). The most abundant TF families up-regulated in wild-type roots were HSFs (24), ERFs (22), NACs (21), MYBs (15), and WRKYs (15) (Figure 2A) while those down-regulated in wild-type roots were bHLHs (11), bZIPs (10), and NF-YAs (8) (Figure 2B).

### 2.2. Heat Stress Regulation of Genes in the Roots of TaHsfC2a-Overexpressing Wheat Seedlings

As indicated above, previously, it has been reported that the *TaHsfC2a*-overexpressing plants exposed to heat stress (2 h of heat treatment at 43 °C) showed significantly better recovery rates (> 90%) in the subsequent 3 weeks following the heat treatment than wild-type plants (15%) [27]. The evidence presented by Hu et al. [27] also indicated that the roots played a role in the recovery phenomenon. To understand the potential mechanisms involved in the recovery of *TaHsfC2a*-overexpressing lines after heat stress, we performed an RNA-seq analysis of the roots. The analysis showed that 1535 and 2487 genes were up- and down-regulated in transgenic roots in response to heat stress (Figure 1; Appendix A).

In the up-regulated gene sets, several GO categories were observed, including secondary metabolic processes and responses to stress, and amine metabolic processes were overrepresented in the wild-type plants (Figure 3), which were also overrepresented in the transgenic line. To further understand the differences in GO enrichment between wild-type and transgenic plants, we have selected highly significant GO categories (FDR ≤ 0.001) from the BLAST2GO enrichment analysis. We found 38 categories over-represented relative to the wild-type in the transgenic up-regulated gene dataset, including the regulation of hydrogen peroxide metabolic process, the regulation of reactive oxygen species metabolic process, glutathione metabolic process, flavonoid biosynthetic process, calcium ion binding, transition metal ion transport, UDP-glucosyltransferase activity, and DNA binding and regulation of the RNA biosynthetic process (Appendix A). A total of 42 GO categories down-regulated in the transgenic roots relative to the heat-treated wild-type, including the nucleosome and chromatin assembly, protein-DNA packaging related, oxidation-reduction process, cell wall polysaccharide, metal ion binding, and membrane transporter activity (Appendix A). Among the DEGs between the control and heat-treated seedlings, 137 TFs were up-regulated, and 128 TFs were down-regulated in the transgenic roots, respectively (Figure 2; Appendix A). The most abundant TF families up-regulated in the roots of *TaHsfC2a*-overexpressing plants were HSFs (22), ERFs (17), C2H2 zinc fingers (13), NACs (12), bZIPs (12), and bHLHs (12) (Figure 2A), while the most abundant TFs down-regulated were bHLHs (17), MYBs (15), NACs (15), and NF-YA (12) (Figure 2B). These results indicated that several TFs were differentially regulated in the transgenic roots.

### 2.3. TaHsfC2a Negatively Regulates Root ROS Levels after Heat Treatment

As indicated above, the analyses found several GO categories related to ROS, such as those involved in the regulation of hydrogen peroxide metabolism, regulation of ROS metabolic processes, and glutathione metabolism, in both the wild-type and *TaHsfC2a*-overexpressing plants (Appendix A). It is well known that heat stress generates ROS [28,29], and class III peroxidases have important functions in maintaining the ROS balance in the plant cell by consuming and/or producing ROS [30]. Therefore, a comparative analysis was performed, and it revealed the differences in peroxidase expressions between wild-type and the transgenic line after heat treatment. At least 70 class III peroxidases were significantly down-regulated in the roots of *TaHsfC2a*-overexpressing plants, while only four peroxidases significantly up-regulated when compared with the wild-type after heat treatment (Table 1). To understand whether the down-regulation of peroxidases in the roots of *TaHsfC2a*-overexpressing plants caused a difference in the accumulation of ROS, hydrogen peroxide levels were measured in the roots of the wild-type and transgenic line before and after heat treatment and found that transgenic roots accumulated lower hydrogen peroxide than wild-type roots (Figure 4). These results indicated that the overexpression of *TaHsfC2a* may be negatively regulating hydrogen peroxide level in transgenic roots by down-regulating the hydrogen peroxide-producing class III peroxidases.

### 2.4. Tahsfc2a-Overexpression Affected Iron Transport Genes and Iron Accumulation in Roots after Heat Treatment

In addition, Fe ion transport and nicotianamine-related GO categories were highly up-regulated in both wild-type and *TaHsfC2a*-overexpressing line transcriptomes after heat stress (Appendix A). Nicotianamine is a precursor of phytosiderophores, which are excreted by the roots of Poaceae species, and mediate the chelation and acquisition of Fe ions [31]. Recently, Distefano et al. [11] identified Fe and ROS-dependent cell death could be induced by heat stress in Arabidopsis roots. To explore a possible involvement of Fe-mediated cell death phenomenon in heat stress tolerance, we examined the expression profiling of genes related to Fe transport and nicotianamine synthase activity and found heat to down-regulate the expression profiles of these genes in *TaHsfC2a* over-expressing plants when compared with wild-type roots (Figure 5). Further, the *TaHsfC2a* over-expressing plant roots were found to accumulate low Fe ions when compared with the wild-type roots after heat treatment (Figure 6). Therefore, these results suggested that *TaHsfC2a*-overexpression was reducing Fe accumulation in the roots, and thus, in turn, reducing ROS- and iron-mediated cell death response, enabling transgenic roots to recover from the detrimental effects of heat stress.

### 2.5. Wheat Root and Leaf Transcriptomes Show Low Correlation under Heat Stress

To explore if wheat roots and leaves/shoots show similar transcriptional responses to heat, we have compared our root transcriptional data to recently published leaf transcriptome data [32]. The cultivars used in these two studies were different; however, the age of the seedlings and the temperature and duration of heat treatment were similar between the two experiments. Overall, the root transcriptome after heat treatment showed relatively low correlations with the transcriptome of 1 h (r = 0.42) and 6 h (r = 0.27) heat-treated leaves when both up- and down-regulated genes were included in the analysis (Figure 7). Furthermore, our analysis also showed that 3331 genes were at least twofold or more differentially regulated (log_2_ FC ≥ 2) in the roots under heat stress. Of these 3331 genes, only 474 (14%) and 546 (16%) genes were significantly differentially regulated in 1- and 6-h in heat-treated leaves, respectively (Appendix A). Surprisingly, 29 and 104 genes showed opposite transcript levels in the 1- and 6-h heat-treated leaves, respectively (Appendix A). In addition, several NAS genes were up-regulated in the roots; hence, we looked at the transcript levels of the nicotianamine synthases (NAS) in the leaves. However, none of the NAS-related genes were significantly up-regulated in the leaves (Table 2). These results indicated that the overall transcriptomic responses of heat-treated leaves and heat-treated roots may not be similar, and root-specific iron-related mechanism may exist under heat stress.

## 3. Discussion

Heat stress is becoming an immense problem in crop production around the world. Although heat can have a drastic effect on roots [33], most studies have focused on above-ground tissues [23]. Here, we have analyzed the effects of heat shock on seedling roots in both the wild-type and a transgenic line over-expressing *TaHsfC2a*. Our analyses showed that heat stress does indeed have a major effect on the root transcriptome. Further, our heat-stressed root transcriptome data were compared with those of the above-ground plant parts reported by Liu et al. [32]. These results showed that although general categories of differentially expressed root genes were like those seen in wheat leaves [32,34], very little correlation, if any, could be found between the actual genes differentially expressed in different tissues (see also below). Organ-specific (roots as compared to leaves) expressions patterns of the *TaHsf* gene family between roots and leaves of wheat in response to heat stress were also reported in wheat [18]. In another study, heat-responsive TF expression patterns were found different in different wheat tissues, such as flag leaves and filling-grains [22]. Similar results have been reported in root hairs and striped roots in soybeans [35]. Furthermore, He et al. [36] showed that transcriptomic responses to heat stress differ between the different tissues of maize (*Zea mays* L.). The alterations of photosynthesis-related proteins in above-ground parts of wheat during heat-stress could be another difference between wheat roots and leaves under heat stress [37]. Another potential difference between wheat roots and leaf transcriptomes under stress is, found by Su et al. [38], the transcriptional regulation of zeatin, brassinosteroid, and flavonoid biosynthesis pathways during heat stress response in leaves [38] but not in the root transcriptome (this study).

The GO enrichment analysis of the DEGs showed that responses to environmental stimuli (heat, light, temperature), ROS, transcriptional regulation, protein-folding, and chromatin remodeling were affected significantly by heat treatment in both the wild-type and transgenic roots. Similar categories were reported from the transcriptome analysis of wheat roots under heat stress [26]. A transcriptome and protein analysis of soybean (*Glycine max* (L.) Merr.) roots under heat also found similar GO categories [35]. The categories related to stress, chromatin, and ROS were also reported in heat-treated wheat seedling leaves [32]. Recently, *TaHsfA6b*-4D was shown to contribute to unfolded protein responses under heat stress to maintain protein homoeostasis in the cells [39]. In tomatoes, HSFA1a plays a key role in chromatin spatial reorganization and enhances the expression of heat-stress responsive genes under heat stress [40]. Together with these studies and our RNA-seq results, it can be suggested that *TaHSFC2a* could be involved in protein homeostasis and chromatin reorganization. In addition, our root transcriptome study identified Fe- and nicotinamide-related GO categories, and similar categories were identified by Luo et al. [26] in wheat roots under heat.

Transcription factors can play a significant role in regulating cellular responses to abiotic stress. The TF families most affected by the heat treatment were HSFs, ERFs, NACs, bHLHs, and MYBs, WRKYs in both wild-type and transgenic roots, and these TFs were also reported in wheat leaves after heat stress [32]. TFs, such as HSFs, ERFs, and WRKYs, were also found in heat-treated soybean roots and maize seedlings [9,35]. When we compared our transcriptome data with the heat-treated leaf transcriptome of wheat [32], we found that G2-like, NF-X1, GRF, RAV, and LBD TF families were differentially regulated only in the roots. G2-like TFs were also differentially expressed in Arabidopsis roots and Brassica rapa whole-plant tissues under abiotic stress conditions [41,42]. *NF-X1* was differentially expressed in Arabidopsis whole-plant tissues under heat stress [43], which is consistent with our transcriptome result on *NF-X1*. The *grf7* Arabidopsis mutants were more tolerant to salinity and drought stress, and GRF was found to regulate the expression of *DREB2A* [44]. *DREB2A* promotes plant survival under severe environmental stress, including high temperatures [44]. At least six *GRFs* were down-regulated in the transgenic line; however, only four were down-regulated in wild-type roots. RAV1 and LBD genes were shown to respond to environmental stimuli in Arabidopsis [45,46]. These TFs were expressed in roots (this study) but not in leaves [32], suggesting that these TFs may be root specific, and may play a role in the heat stress tolerance of wheat.

Several heat shock proteins, including *HSP70* and *HSP90*, were highly up-regulated after heat treatment in both the wild-type and transgenic roots, indicating a strong transcriptional response of root tissues (Appendix A). In wheat, the RNA-seq of seedling leaves after heat treatment increased the expression of several *Hsf* (class A, B and C), and *HSP* genes [47]. The Hsf TFs play a crucial role under heat stress by regulating HSPs and other stress-related genes. Previous studies have reported that the overexpression of *HsfA* induced the expression of a large array of stress-related genes [48,49,50]. The overexpression of *TaHsfA2-1* and *TaHsfA2e-5D* in Arabidopsis was found to up-regulate *HSP* genes and confers stress tolerance [51,52]. HsfC2 members are monocot specific [19], and their effects on the whole transcriptome have not been studied before. Our results showed that *TaHsfC2a* overexpression affects a large array of genes under heat stress in roots when compared with wild-type plants. Although several studies observed that the overexpression of *Hsfs* increased the survival of plants and conferred stress tolerance [27,53], the underlying molecular mechanisms are unknown. For example, the overexpression of *TaHsfA2-1, TaHsfA2e-5D*, and *TaHsfA2-7-AS* in Arabidopsis enhanced heat tolerance [51,52,54]. However, the molecular mechanisms are yet to be studied.

Here, through transcriptional profiling, we aimed to decipher the mechanism behind the survival of *TaHsfC2a*-overexpressed transgenic plants after heat stress as reported by Hu et al. [27]. We found that a group of peroxidases were down-regulated in transgenic lines when compared with wild-type plants after heat treatment. The down-regulation of hydrogen peroxide-producing peroxidases would decrease the hydrogen peroxide accumulation in the transgenic roots. This may be one of the reasons for the increased survival of *TaHsfC2a*-overexpressing plants as increased ROS levels could trigger cell death in roots. Previous studies showed that the peroxidases gene expression possibly affected the level of hydrogen peroxide in Arabidopsis roots [55,56]. This view is consistent with the observation of the reduced accumulation of hydrogen peroxide in the transgenic roots in this study, as class III peroxidases are involved in generating hydrogen peroxide [57,58]. Furthermore, a recent study in hair grass (*Agrostis scabra* Willd.) showed that the heat-tolerant grass accumulated lower ROS levels in the roots than in the heat-susceptible creeping bent grass (*A. stolonifera* L.) under heat stress [59].

In addition to peroxidases, nicotianamine-related transcripts were less abundantly expressed in the roots of *TaHsfC2a*-overexpressing plants in response to heat when compared with wild-type heat responses. Nicotianamine is a precursor of phytosiderophores, which are important for the iron uptake by the Poaceae family plants, including wheat [31]. Using a chelation-based strategy to form soluble Fe (III) complexes, which are then taken up by the roots [60]. Recently, Distefano et al. [11] identified ferroptosis-like cell death in Arabidopsis roots in response to heat stress that is dependent on ROS and iron accumulation. Therefore, our results suggest that reduced ROS and iron levels may be critical factors contributing to the survival of *TaHsfC2a*-overexpressing plants after heat treatment through the reduction of ferroptotic cell death. Ferroptosis-like cell death has been identified in diverse species with distinct features of the disruption of iron and ROS-homeostasis [61]. Ferroptosis is an ancient evolutionary mechanism, which is conserved across animal and plant species [11]. In animal cell culture studies, the knockdown of HSF1 was found to enhance ferroptosis through the accumulation of intracellular iron and the associated lipid peroxidation in cancer cells and xenograft models [62], which indicates that *TaHsfC2a* plays a key role in ferroptosis in plants as both genes belong to the Hsf gene family.

## 4. Materials and Methods

### 4.1. Plant Materials and Growth Conditions

The spring wheat (*Triticum aestivum* L.) cultivar (cv.) Fielder (wild-type/untransformed control) and *TaHsfC2a-B* overexpressing transgenic line (C2a-17) in cv. Fielder background were described in Hu et al. [27]. Seeds were washed, kept on wet paper towel at 4 °C for 5 days, and then transferred to 12 °C to induce germination. Two days after germination, the seedlings retained within the moistened paper towel were transferred to room temperature for 1 day, and then transferred to plastic containers containing sterilized reverse osmosis water and allowed to grow for one further day at room temperature. Then, the 4-day-old seedlings were placed into a Hoagland and Arnon nutrient solution No.2 [63] containing plastic container and transferred to a controlled-environment facility (CEF; CSIRO, St. Lucia, Queensland, Australia), where the environmental factors, such as photoperiod, temperature, light, and relative humidity can be controlled. The seedlings were kept under a 16/8-h (day/night) thermoperiod of 22/16 °C with a matching photoperiod (500 μmol m^−2^s^−1^) and 60 to 80% relative humidity for a further 3 days in the CEF. Seven-day-old seedlings in the container with Hoagland and Arnon nutrient solution No.2 were kept in a water bath at 43 °C. Once the nutrient solution reached 43 °C, the seedlings were heat-treated for 2 h at 43 °C, while untreated seedlings were kept in the CEF as control samples. The heat-treated and untreated seedlings were then kept in the CEF for one further day before being sequenced.

### 4.2. RNA Preparation and Sequencing

Eight-day-old seedling roots from the wild-type and C2a-17 lines were collected from heat-treated and untreated plants. Root tissues were immediately frozen in liquid nitrogen and then stored at −80 °C for further use. Roots from at least six seedlings, from each container with nutrient solution, were pooled to make up one replicate. Total RNA was extracted from the pooled root tissues, and per replicate using RNeasy Plant Mini Kit (Qiagen, Melbourne, Victoria, Australia) according to the manufacturer’s instructions. Four biological replicates (biologically distinct samples from same treatment) per treatment were used for each of the treated and control samples of wild-type and overexpressing plants in this study. Nucleic acid quantity was measured with a NanoDrop ND-1000 UV-Vis Spectrophotometer (Nano Drop Technologies, Wilmington, DE, USA), and quality was determined using Agilent 2100 Bioanalyzer (Agilent Technologies, CA, USA).

A TruSeq stranded messenger RNA (mRNA) kit was used to generate 100 base pair (bp) paired-end libraries, according to the manufacturer’s protocol (Illumina Inc., San Diego, USA). Libraries were barcoded prior to sequencing. Sequencing was performed using an Illumina HiSeq 2500 platform with four lanes, and all samples were run on each lane which made up four technical replicates (same sample run on four lanes) per sample. Library preparation and sequencing were performed at the Australian Genome Research Facility (AGRF; Melbourne, Victoria, Australia). Sequence files were deposited to the National Centre for Biotechnology Information (NCBI) Sequence Read Archive under bioproject PRJNA498129.

### 4.3. RNAseq Analysis

We used the wheat reference genomic sequence of cv Chinese Spring, RefSeq v1.0 [64], and the Tuxedo package, previously described [65], to detect genes that were differentially expressed between the control and heat-treated seedlings. Firstly, we excluded base-call errors arising during the sequencing process using the software package SolexaQA. RNA-seq reads were trimmed so that all remaining bases had a PHRED score > 30 and their final read length was at least 70 bp with both right and left reads fulfilling these requirements. Filtered and trimmed paired reads were aligned to the Chinese Spring cv. RefSeq v1.0 annotation (files accessed from URGI on 13 March 2017) using Tophat2 v2.1.1 with Bowtie2 v2.2.9 as the aligner. Binary alignment maps (BAM files) were produced using the program SAMtools v1.3.1. Transcript fragments were assembled and normalized to yield fragments per kb per million reads using the package Cufflinks v2.2.2, prior to replicate concatenation by Cuffmerge, and a differential gene expression calling by Cuffdiff. A false discovery rate (FDR) and multiple comparison correction (Bonferroni correction) were performed within the Cuffdiff process, identifying which genes were significantly differentially expressed between the control and heat-treated seedlings for each genotype, the wild-type Fielder, and the *TaHsfC2a*-overexpressing line. Differentially expressed genes (DEG) were then filtered from each dataset using FDR adjusted *p* value < 0.05 and > log_2_ twofold change cut-off values and divided into up- and down-regulated sets based on the positive or negative fold change values, respectively. Venn diagrams were produced using these filtered datasets with the aid of a webtool produced by the Bioinformatics and Evolutionary Genomics group based at the University of Ghent (http://bioinformatics.psb.ugent.be/webtools/Venn/) (accessed on 1 December 2019).

### 4.4. Gene Ontology Enrichment Analysis

The set of high confidence coding sequences were identified from our RNA-seq. Then, these sequences were annotated using the BLAST2GO [66] server based at CSIRO Agriculture and Food. This produced set of ca.138,000 annotated genes and was used as a background reference to define the wheat GO categories for the enrichment analysis. The up- and down-regulated genes (≥2 log_2_-fold and ≤−2 log2-fold) were used as separate test inputs. Standard parameters were applied for the BLASTx, mapping, and annotation steps and for functional enrichment testing using the Fisher’s exact test module (FDR adjusted *p* value < 0.05). Highly significant GO terms (FDR; *p* value < 0.001) are presented in Appendix A. The heatmaps incorporated in this study were generated using the ggplot2 package (tidyverse 1.2.1) in R. We have used the BiNGO plugin for Cytoscape to identify significantly overrepresented GO categories from the wild-type, and used custom wheat GO term annotation as a reference (http://www.psb.ugent.be/cbd/papers/BiNGO/ [67]). The differentially expressed gene sequences were used in PlantTFDB4.0 to identify putative TFs [68].

### 4.5. Hydrogen Peroxide Measurement Using Hydroxyphenyl Fluorescein (HPF)

At least 15 heat-treated and untreated roots from 8-day-old seedlings (similar treatments were applied as described for RNA sample collection) were incubated for 2 min in 0.1 M phosphate buffer, pH 7.4, containing 5 µM 3′-(p-hydroxyphenyl) fluorescein (Sigma, St. Louis, CA, USA) [69]. Roots were then mounted on a glass slide in a drop of buffer, covered, and observed under a fluorescence microscope (Leica MZ16FA) with excitation/emission of 480/510 nm. Fluorescence images of roots were immediately captured from roots, with the microscope setting unaltered for all the treatments. ImageJ was used to measure the staining intensity of the fluorescence images.

### 4.6. Histochemical Detection of Fe^3+^

The histochemical study for the detection of Fe was carried out following the previous methods with slight modifications [70,71]. One-week-old seedlings in a container containing Hoagland and Arnon nutrient solution No.2 were exposed to heat stress in a water bath at 43 °C for 2 h, as described for the RNA sample collection, and then incubated in 7% potassium ferrocyanide and 3% HCl (1:1 *v*/*v*) for 15 h at room temperature. Roots were washed three times with running distilled water and samples collected from the root hair zone. The sections were mounted onto slides, stained with Prussian blue, covered with a cover slip, and observed under the microscope. Blue pigmentation in tissues caused by the presence of ferric ferrocyanides, formed by an interaction between Fe ions and ferrocyanides, was observed.

## 5. Conclusions

In conclusion, our current study showed that *TaHsfC2a* is modulating ROS and iron levels in response to heat stress in wheat roots. Consequently, this study found the existence of ferroptosis-like cell death in wheat, and *TaHsfC2a* could be a key player in this mechanism. Further studies in this area are highly warranted to test the utility of *TaHsfC2a* overexpression in developing thermotolerant varieties in wheat.

## Figures and Tables

**Figure 1 ijms-24-03099-f001:**
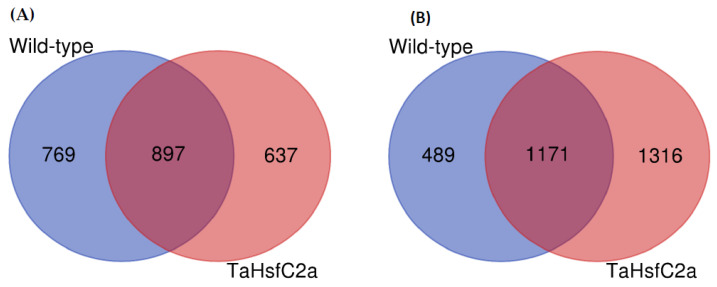
Differentially regulated genes in the wild-type and the transgenic roots under heat stress. (**A**) upregulated genes (**B**) downregulated genes. Wild-type (control v heat) and *TaHsfC2a* (control v heat). Fold Change log_2_ >2; False Discovery Rate corrected *p* < 0.05.

**Figure 2 ijms-24-03099-f002:**
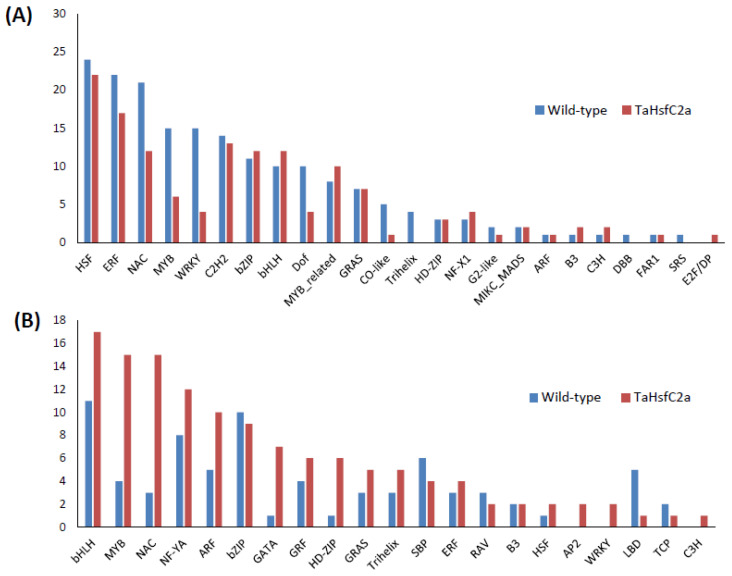
Number of transcription factors identified in the wild-type and the transgenic roots in response to heat stress (**A**) upregulated genes (**B**) downregulated genes. Wild-type (control v heat) and *TaHsfC2a* (control v heat).

**Figure 3 ijms-24-03099-f003:**
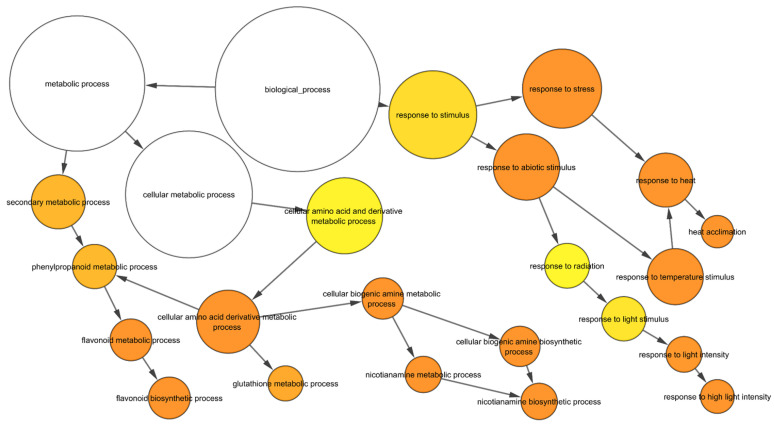
GO enrichment of upregulated genes in response to heat in wild-type. The yellow to orange colour of the circles correspond to the level of significance of the overrepresented GO categories. The size of the circle is proportional to the number of genes in the category. Benjamini & Hochberg false discovery rate corrected *p* value < 0.05.

**Figure 4 ijms-24-03099-f004:**
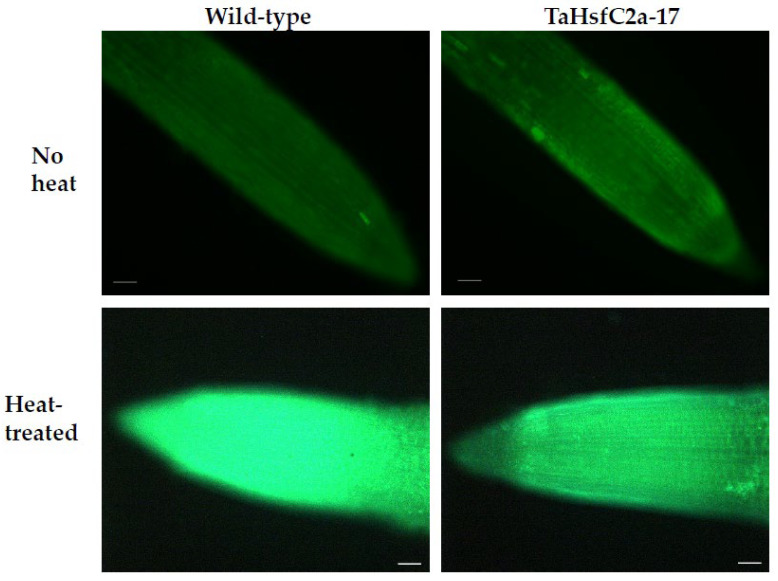
Hydrogen peroxide along the roots of wild-type and transgenic line with and without heat treatment. Roots of eight-day-old wheat plants were stained with 3′-(phydroxyphenyl) fluorescein (HPF) for hydrogen peroxide. Scale bar, 100 µm.

**Figure 5 ijms-24-03099-f005:**
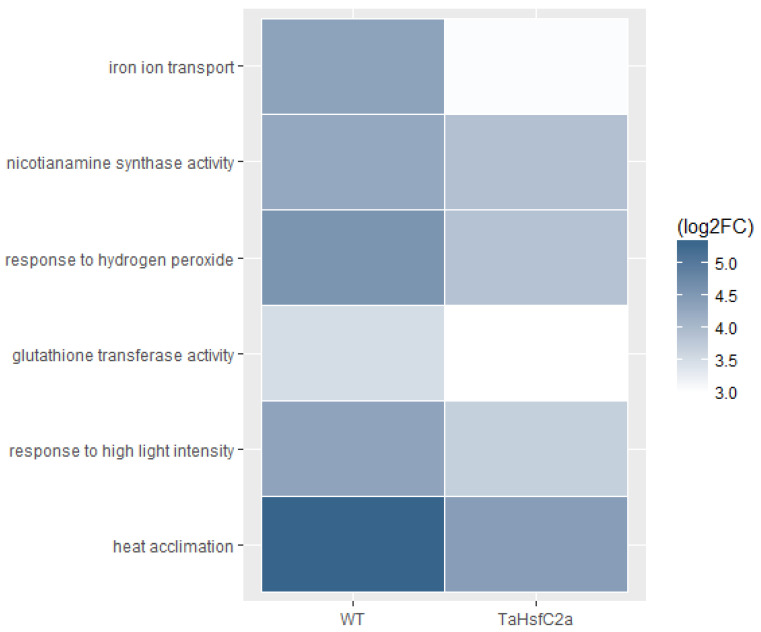
Transcripts from iron transport, nicotianamine synthase, reactive oxygen species and heat stress GO categories were low abundant in the transgenic roots. WT, wild-type (control v heat); *TaHsfC2a* (control v heat); log_2_ FC, Fold Change log_2_ ≥ 2.

**Figure 6 ijms-24-03099-f006:**
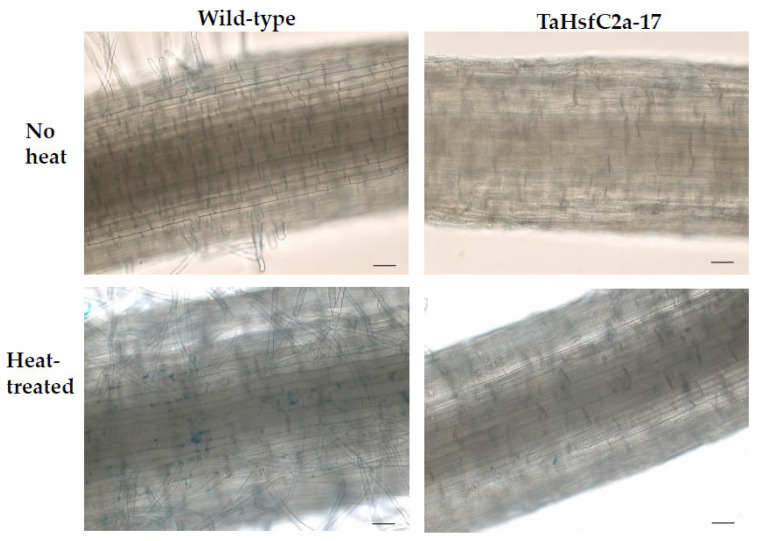
Iron accumulation along the roots of wild-type and transgenic line with and without heat treatment. Roots of eight-day-old wheat plants were stained with Prussian blue for ferric ions. Scale bar, 100 µm.

**Figure 7 ijms-24-03099-f007:**
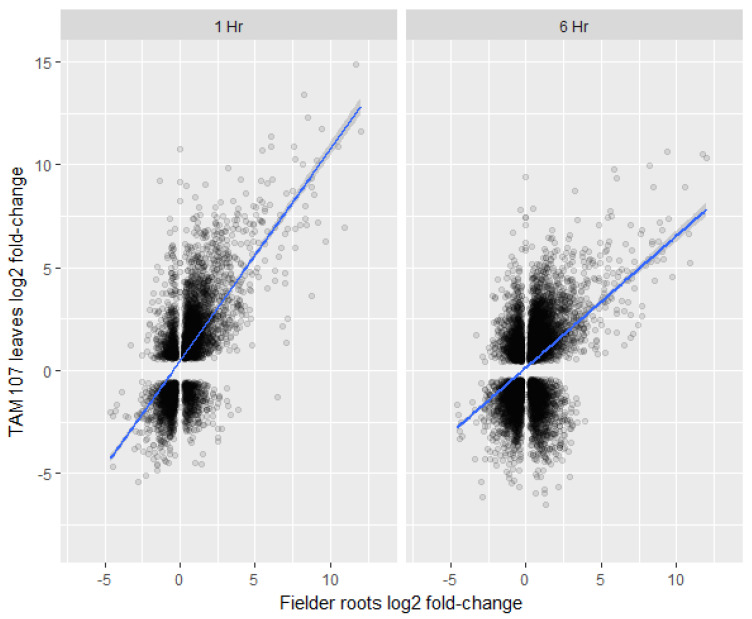
Comparison of differentially expressed gene log fold-changes between Fielder roots (wild-type) (*x*-axis) 2 h post heat treatment and TAM107 leaves (*y*-axis) 1 h post heat treatment (left panel) or 6 h post heat treatment (right panel). Significant Kendall’s rank correlations of 0.42 and 0.27 were calculated for 1 hour and 6 h datasets respectively.

**Table 1 ijms-24-03099-t001:** Differential expression of class III peroxidases in the roots of *TaHsfC2a* overexpressing plants compared with the wild-type after heat treatment.

Gene ID	Wild-Type_Heat vs. TaHsfC2a_Heat (log_2_ Fold-Change)	Gene ID	Wild-Type_Heat vs. TaHsfC2a_Heat (log_2_ Fold-Change)
TraesCS7D01G369900	−2.0176	TraesCS1D01G080400	−0.70037
TraesCS2B01G616700	−1.87421	TraesCS1A01G318500	−0.70025
TraesCS7A01G452900	−1.56436	TraesCS7D01G084900	−0.69508
TraesCS7A01G345000	−1.52138	TraesCS7D01G369400	−0.68257
TraesCS3A01G180100	−1.41352	TraesCS4A01G076600	−0.64204
TraesCS1D01G402200	−1.39785	TraesCS7D01G212900	−0.63378
TraesCS5A01G400500	−1.31626	TraesCS7B01G219900	−0.6316
TraesCS7B01G251100	−1.24972	TraesCS6B01G071200	−0.61914
TraesCS5B01G040900	−1.15713	TraesCS7A01G474200	−0.56937
TraesCS4A01G196000	−1.155	TraesCS4B01G232700	−0.56258
TraesCS5B01G405300	−1.13893	TraesCS7D01G315600	−0.55323
TraesCS2B01G560500	−1.12414	TraesCS4D01G233900	−0.55036
TraesCS7D01G334400	−1.06495	TraesCS1B01G115800	−0.54597
TraesCSU01G113200	−1.01216	TraesCS2B01G614400	−0.54581
TraesCS7D01G347300	−1.009	TraesCS4B01G118900	−0.52509
TraesCS6A01G118300	−0.99397	TraesCS7A01G453000	−0.52401
TraesCS2A01G573900	−0.97832	TraesCS7B01G325800	−0.51675
TraesCS7A01G094600	−0.96703	TraesCS7A01G319100	−0.49718
TraesCS4B01G347700	−0.9383	TraesCS4A01G389000	−0.49609
TraesCS5A01G148600	−0.93731	TraesCS7D01G461200	−0.47394
TraesCS2A01G509800	−0.91224	TraesCS6B01G333800	−0.46766
TraesCS1B01G115700	−0.89965	TraesCS3B01G149000	−0.45068
TraesCS3A01G297100	−0.89468	TraesCS7D01G334300	−0.44337
TraesCSU01G137300	−0.87359	TraesCS4A01G196400	−0.42716
TraesCS1B01G096600	−0.87081	TraesCS7A01G424100	−0.42195
TraesCS2B01G614100	−0.83599	TraesCS3B01G210000	−0.42106
TraesCS5B01G147200	−0.82952	TraesCS7B01G274200	−0.41739
TraesCS1B01G096800	−0.81718	TraesCS1D01G096300	−0.4071
TraesCS6A01G138000	−0.81112	TraesCS7D01G417400	−0.4032
TraesCS2B01G284900	−0.80031	TraesCS2A01G573700	−0.39615
TraesCS7A01G339700	−0.7802	TraesCS7B01G353200	−0.34837
TraesCS2B01G494800	−0.77567	TraesCS1A01G104200	−0.34208
TraesCS1A01G319000	−0.77424	TraesCS7D01G442300	−0.3336
TraesCS5D01G410500	−0.76918	TraesCS2D01G451200	0.459274
TraesCS2B01G613900	−0.75796	TraesCS7B01G298400	0.484196
TraesCS2B01G124600	−0.72275	TraesCS2B01G124300	0.615129
TraesCS7B01G118200	−0.71052	TraesCS3D01G305000	0.778914

**Table 2 ijms-24-03099-t002:** Fold changes (heat/control ratio) observed at the transcript levels of nicotianamine synthase genes in the roots and the leaves of the wheat cultivars, wild-type (this study) and TAM107 [32], respectively.

Gene ID	Gene Description	Wild-Type Roots 2 h (Control vs. Heat) Fold Change	TAM107 Leaves 1 h (Control vs. Heat) Fold Change	TAM107 Leaves 6 h (Control vs. Heat) Fold Change
TraesCS6B01G186100	nicotianamine synthase	6.02715	ns	ns
TraesCS6A01G165200	nicotianamine synthase	5.57004	ns	ns
TraesCS6D01G148600	nicotianamine synthase	5.32415	ns	ns
TraesCS3B01G068500	nicotianamine synthase	5.04746	ns	ns
TraesCS6A01G386200	nicotianamine synthase	4.81333	ns	ns
TraesCS6A01G093000	nicotianamine synthase	4.64126	ns	ns
TraesCS2D01G049200	nicotianamine synthase	4.48658	ns	ns
TraesCS4D01G184900	nicotianamine synthase	4.45258	ns	ns
TraesCS2B01G060800	nicotianamine synthase	4.32076	ns	ns
TraesCS6B01G187400	nicotianamine synthase 1	4.3186	ns	ns
TraesCS6A01G165100	nicotianamine synthase	4.31789	ns	ns
TraesCS2A01G049900	nicotianamine synthase	4.28389	ns	ns
TraesCS6B01G186000	nicotianamine synthase	4.16596	ns	ns
TraesCS5A01G552000	nicotianamine synthase	4.165	ns	ns
TraesCS3B01G068400	nicotianamine synthase	4.11312	ns	ns
TraesCS6B01G425200	nicotianamine synthase	4.02728	ns	ns
TraesCS4B01G183900	nicotianamine synthase	3.86027	ns	ns
TraesCS5A01G552400	nicotianamine synthase	3.76404	ns	ns
TraesCS6D01G370800	nicotianamine synthase	3.64214	ns	ns
TraesCSU01G125500	nicotianamine synthase	3.16693	ns	ns
TraesCSU01G125100	nicotianamine synthase	3.0556	ns	ns
TraesCS6D01G382900	nicotianamine synthase 3	2.40638	ns	ns
TraesCS2D01G033000	nicotianamine synthase 2	1.20994	ns	ns
TraesCS4B01G184000	nicotianamine synthase6	1.0342	ns	ns
TraesCS2A01G033500	nicotianamine synthase-like 5 protein	0.862108	ns	ns
TraesCS4A01G120900	nicotianamine synthase6	0.773459	ns	ns
TraesCS4D01G185100	nicotianamine synthase6	0.538083	ns	ns
TraesCS2B01G111100	nicotianamine synthase 3	ns	ns	−1.67492
TraesCS2D01G094200	nicotianamine synthase 3	ns	ns	−2.01409

ns, not significant.

## Data Availability

All sequence files were deposited to the National Centre for Biotechnology Information (NCBI) Sequence Read Archive (SRA) under bioproject PRJNA498129.

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
