# Peer review of "Transcriptome Analysis of *Heat Shock Factor C2a* Over-Expressing Wheat Roots Reveals Ferroptosis-like Cell Death in Heat Stress Recovery"

_ijms, 2023, doi:10.3390/ijms24043099_

Round 1
Reviewer 1 Report
use the scientific names as italic.
why did not the gene expression analysis for confirm the results?
Author Response
Comments and Suggestions for Authors
use the scientific names as italic. –
Response from authors: Thanks for this comment. We noticed that while copy/pasted on the template of IJMS had changed format in several places. Now, it has been modified properly.
why did not the gene expression analysis for confirm the results?
Response from authors: Thanks for notifying this point. As several marker genes (especially Hsf and HSP genes) were observed across the treatments/replicates, authors thought it is obvious, so it has been mentioned in the paper and provided in supplementary files.
English language checked throughout the paper.
Reviewer 2 Report
Kalaipandian et al. have presented a nice study on transcriptome analysis of Heat Shock Factor C2a over-expressing wheat roots. The topic is of interest and can be published. For me, the only requirement is that there are a number of studies on heat stress in wheat recently, specifically in 2022. These studies should be added to the manuscript. Moreover, please add the novelty and future prospects of the study to the abstract. After implementing these minor suggestions, I do believe that the manuscript can be accepted.
Author Response
Comments and Suggestions for Authors
Kalaipandian et al. have presented a nice study on transcriptome analysis of Heat Shock Factor C2a over-expressing wheat roots. The topic is of interest and can be published. For me, the only requirement is that there are a number of studies on heat stress in wheat recently, specifically in 2022. These studies should be added to the manuscript. Moreover, please add the novelty and future prospects of the study to the abstract. After implementing these minor suggestions, I do believe that the manuscript can be accepted.
Authors response: Thanks to the reviewer for providing valuable comments to improve the manuscript. We have included 6 new references, and also included new points in the abstract.